# Effects of horticultural therapy versus handiwork on anterior cingulate cortex activity in people with chronic low back pain: A randomized, controlled, cross-over, pilot study

**Alexandra Rören**[1,2,3,4]*, **Clement Debacker**[5,6], **Marc Saghiah**[6], **Catherine Bedin**[3], **Anna Fayolle**[5,6], **Hendy Abdoul**[7], **Marie-Martine Lefèvre-Colau**[2,3,4,8], **François Rannou**[3,4,8,9], **Catherine Oppenheim**[5,6], **Christelle Nguyen**[3,4,8,9]

1 Département des Sciences de la Rééducation et de la Réadaptation, Faculté de Santé, Université Paris Cité, Paris, France, 2 INSERM UMR 1153, Centre de Recherche en Épidémiologie et Statistique Sorbonne Paris Cité, Université Paris Cité, Paris, France, 3 AP-HP, Centre-Université de Paris, Service de Rééducation et de Réadaptation de l'Appareil Locomoteur et des Pathologies du Rachis, Hôpital Cochin, Paris, France, 4 Fédération pour la Recherche sur le Handicap et l'Autonomie, Paris, France, 5 GHU-Paris Psychiatrie et Neurosciences, Hôpital Sainte Anne, Paris, France, 6 Institute of Psychiatry and Neuroscience of Paris (IPNP), INSERM U1266, IMA-Brain Team, Université Paris Cité, Paris, France, 7 Direction de la Recherche Clinique et de l'Innovation de l'AP-HP, Hôpital Cochin, Paris, France, 8 Faculté de Santé, UFR de Médecine, Université Paris Cité, Paris, France, 9 INSERM UMR-S 1124, Toxicité Environnementale, Cibles Thérapeutiques, Signalisation Cellulaire et Biomarqueurs, Campus Saint-Germain-des-Prés, Université Paris Cité, Paris, France

* alexandra.roren@aphp.fr

**Data Availability Statement:** Data are owned by the promotor Assistance-Publique Hôpitaux de Paris (AP-HP). Data cannot be shared publicly

## Abstract

To assess the efficacy of horticultural therapy (HT) on anterior cingulate cortex (ACC) activity and the changes in rumination and catastrophizing scores in individuals with chronic low back pain (LBP). We conducted a randomized, controlled, cross-over, 3-week pilot study (ClinicalTrials.gov Identifier: NCT04656158). The departments of physical medicine and rehabilitation (hospital grounds and occupational therapy room) and imaging research were involved. The participants were adults with non-specific chronic LBP. All participants underwent two 90-min HT sessions and two 90-min handiwork sessions per week. The activity sequence order was randomized, and the activities were separated by a wash-out period of 1 week. Each participant underwent 3 brain MRIs: before, after the first, and after the second activity. The primary outcome was the change in ACC perfusion in ml/100g/min using arterial spin labeling MRI. The secondary outcomes were the changes in self-reported rumination and catastrophizing scores after each activity compared to baseline. Sixteen participants were included: 14 women (87.5%), LBP intensity (numeric rating scale) mean (SD) 45.1 (27.2)/100, specific activity limitation (Roland Morris disability questionnaire) 9.3 (4.1)/24. Change in ACC perfusion from baseline was -0.1 (10.7), 95% CI [-5.6, 5.8] ml (blood)/100g (tissue)/min after handiwork and -0.1 (8.7), [-4.7, 4.6] after HT and did not differ between the 2 activities (p = 0.91). Change in rumination [-0.5 (4.4) after handiwork and -0.3 (2.8) after HT] and catastrophizing scores [-2 (2.8) after handiwork and -1.4 (2.3) after HT]

because of AP-HP sharing data policy. Data are available from the AP-HP Institutional Data Access (contact via Unité de Recherche Clinique (URC) Necker-Cochin laetitia.peaudecerf@aphp.fr, for researchers who meet the criteria for access to confidential data. All Data not provided in the article may be shared at the reasonable request of any qualified investigator for purposes of replicating procedures and results.

**Funding:** AR Fonds MSERI hôpitaux AP-HP. Centre, Université Paris Cité 2019. AR Fondation Roi Baudoin, (Fonds Luc), 2018 Belgium AR Trophées patients APHP The funders had no role in the design and conduct of the study; collection, management, analysis, and interpretation of the data; preparation, review, or approval of the manuscript; and decision to submit the manuscript for publication.

**Competing interests:** The authors have declared that no competing interests exist.

**Abbreviations:** ACC, anterior cingulate cortex; ASL, arterial spin labeling; CBF, cerebral blood flow; HT, horticultural therapy; LBP, low back pain; PMR, physical medicine and rehabilitation.

did not differ between activities (p = 0.99 and 0.22, respectively). Limited exposure to the interventions and the sample profile (moderate levels of pain) may explain our results. Our results highlight the need for future studies using the most appropriate outcomes to determine the exact effects of nature experiences in people with chronic musculoskeletal disease.

## Introduction

The anterior cingulate cortex (ACC) is part of the neuromatrix of pain and is involved in the emotional valence of pain [1]. The ACC is activated in the case of chronic pain and may contribute to pain persistence [2]. The activity of the ACC is related to rumination, defined as the intrusion of involuntary thoughts that are difficult to "turn off" [3, 4]. A randomized controlled trial by Bratman et al. (2015) showed that both neural activation of the subgenual part of the ACC and rumination scores decreased significantly in 38 young asymptomatic individuals after a 90-minute walk in a natural setting [5].

Chronic low back pain (LBP), defined as pain persisting longer than 3 months is a common condition worldwide that limits activity and is associated with significant costs [6, 7]. Physical exercise improves pain and function in people with chronic LBP [8].

Gardening is a global physical activity involving contact with nature. When used for therapeutic purposes in people with mental, physical or social disability, supervised gardening is called horticultural therapy (HT) [9]. A recent umbrella review reported the positive impact of HT on the mental and physical well-being of people with long-term physical conditions [10]. HT seems well suited to the physical treatment of chronic LBP, as gardening activities involve spinal flexibility, spinal and limb muscle strengthening, and proprioceptive adjustments. HT meets well the criteria for long-term practice as it is associated with pleasure and creativity and can be performed at all stages of life [11].

Few studies have been conducted on the impact of HT on chronic LBP. A non-randomized controlled study of people with chronic pain (fibromyalgia and chronic LBP) found significantly larger improvements in health status, anxiety and coping strategies in the HT group (addition of 7 HT sessions to a standardized pain management program) than in the control group [12].

The main objective of this study was to compare the efficacy of supervised HT sessions with handiwork on change in ACC activation in people with chronic LBP using arterial spin labeling (ASL) perfusion MRI [5]. The secondary objectives were to assess the effect on rumination and catastrophizing scores.

We hypothesized that HT sessions would decrease ACC activation significantly more than handiwork sessions.

## Methods

### Design

We conducted a single-center, randomized, controlled, cross-over, open pilot study over a 3-week period (ClinicalTrials.gov Identifier: NCT04656158).

We reported our study in accordance with the Consolidated Standards of Reporting Trials (CONSORT) checklist [13] (S1 Fig) and the Template for Intervention Description and Replication (TIDieR) [14] (S2 Fig).

Substantial changes were made to the study protocol after the trial commencement to facilitate participant inclusion: recruitment was widened from people on the waiting list for the functional restoration program in our physical medicine and rehabilitation (PMR) department by the use of additional recruitment aids (posters and leaflets). In addition, the inclusion period was extended by 6 months (total duration: 12 months). We did not change any outcomes after the trial had begun.

## Ethics statement

The study was approved by the Comité de Protection des Personnes SUD-EST IV, (n° 19.12.17.40700). All participants provided signed informed consent.

## Blinding

The principal investigator (AR), the occupational therapist who supervised the activities (handiwork and HT) (CB), the radiology technician (AF) and the participants were not blinded to activity sequence allocation. The research engineers (CD and MS) and the data analyst (HA) were blinded.

## Setting and participants

The study was conducted in the physical medicine and rehabilitation (PRM) department of Cochin Hospital and the imaging research department of Saint-Anne Hospital (Paris, France). HT took place on the hospital grounds, and handiwork in the occupational therapy room of the PMR department.

Participants screened for eligibility were scheduled for the baseline face-to-face visit with the principal investigator (AR), a physiotherapist with 20 years of experience in the rehabilitation of people with chronic LBP and 10 years of experience in research.

The main inclusion criteria were 1) age $\geq$ 18 years, 2) non-specific chronic LBP, 3) indication for in-patient rehabilitation, 4) up-to-date tetanus vaccination and 5) being able to walk 2km. The main non-inclusion criteria were 1) contraindications to MRI, 2) sick leave for chronic LBP $\geq$ 3 months in the last year and 3) not in paid employment.

## Study protocol

**Randomization.** Randomization was centralized, and participants were randomly allocated in a 1:1 ratio to one of the 2 sequences of activities: handiwork-then-HT or HT-then-handiwork. An independent statistician provided a computer-generated randomization list with permuted blocks of 4 participants.

**Intervention.** HT and handiwork activities have been part of the routine care of people with chronic LBP in our PMR department for many years (25 years for handiwork and 4 years for HT). Both activities are supervised by experienced occupational therapists with more than 25 years of experience in chronic LBP and more than 15 years of experience in the management of therapeutic workshops. HT was initiated by the individuals who attend the PMR department themselves. The handiwork activity was chosen as a comparator because of its comparable approximate energy expenditure value [15], and because during handiwork in the occupational therapy room, participants had no contact (even visual) with nature.

The dose of 2x90 min of activity was chosen in accordance with the dose of nature exposure used by Bratman et al., (2015) [5] to measure variation in ACC blood perfusion and with the recommendations for physical activity for adults with chronic conditions from the World Health Organization.

Each participant performed 2, 90-minute HT sessions (activity n˚1) and 2, 90-minute handiwork sessions (activity n˚2) supervised by the occupational therapist (CB). The 2 sessions of each activity were performed in the same week, interspersed with 1 week without either activity [16]. Thus, the study lasted for 3 week (S3 Fig).

Participants were asked not to change their lifestyle and usual activities during the study.

The tasks performed during HT sessions included transporting gardening tools, weeding, digging, planting and watering (S4 Fig). The tasks performed during the handiwork activity were transporting wooden boards, measuring, cutting wood with a hand saw, and assembling wood pieces (S5 Fig).

Each participant underwent 3 brain MRI scans: 1 before beginning the activities (reference MRI), 1 immediately after the second session of activity n˚1 and the last immediately after second session of activity n˚2.

**Outcomes.** The primary outcome was the change in ACC perfusion in ml (blood)/100g (tissue)/min after each activity compared to baseline as previously described [5].

The secondary outcomes were the change in self-reported rumination and catastrophizing scores after each activity compared to baseline. The rumination score was one of the 2 scales of the Rumination Reflection Questionnaire (0–60, 0: no rumination, 60: maximal rumination) [17], the catastrophizing score was the sub-score of the Coping Strategy Questionnaire (0–20, 0: no catastrophizing 20: maximal catastrophizing) [18].

Acceptability and satisfaction were collected after each activity using a self-administered numerical rating scale (0 = not acceptable and 100 = maximal acceptability; 0 = minimal satisfaction and 100 = maximal satisfaction).

Safety outcomes were recorded after the first activity, and after the "wash out" period and by using an open-ended question ("Did you have any adverse events?").

**Image acquisition.** Image acquisition and processing were determined according to the study by Bratman et al., (20015) [5]. Scans were acquired on a 3T whole-body MRI scanner (Vantage Galan 3T / XGO; Canon Medical Systems Corporation, Tochigi, Japan) with a 32-channel head coil. The T1-weighted MR images included 160 1.0-mm thick slices with an in-plane resolution of 1 mm$^2$. High-resolution image acquisition was followed by a pseudo-continuous ASL sequence (pCASL) with a readout of 3D single shot Turbo Spin Echo and a postlabel delay (PLD) of 1800 ms; TR = 4910 ms; TE = 12.3 ms; FOV = 195 x 195 mm; matrix size = 64 x 64; 25 axial slices; slice thickness = 4.0 mm; voxel dimensions 3.0469 x 3.0469 x 4.0 mm, 4 averages, for a total acquisition time of 7 min and 52 s.

**Image processing.** The cerebral blow flow quantification equation calculation yielded voxelwise quantitative maps reflecting milliliters of blood per 100 g tissue per minute (volume x time/mass). We coregistered arterial spin labeling (ASL) volumes to each individual's anatomical scans and then performed a combined affine and nonlinear warping process of the anatomical data to standard space, using ASLprep (ASLprep) [19]. We applied the same warping parameters to the ASL data.

We acquired perfusion-weighted data and proton density (PD) maps and then combined information from those per the standard CBF equation quantification [20].

$$CBF = 6000 \times \lambda \frac{exp\left(\frac{PLD(s)}{T_{1b}(s)}\right)}{2T_{1b}(s)\left\{1 - exp\left[-\frac{LT(s)}{T_{1b}(s)}\right]\right\}\varepsilon}\left(\frac{PW}{PD}\right)$$

T1b is T1 of blood and is assumed to be 1.6 s at 3 T (10). λ (0.9 ml/g) is the brain/blood partition coefficient, ε (0.6) is overall efficiency, a combination of inversion efficiency (0.8) and

background suppression efficiency (0.75). PLD (1.8s) is the post-labeling delay used for the ASL sequence, and LT (1.8s) is the labeling duration.

**Statistical analysis.**    Following the results of a study reporting ACC blood perfusion in asymptomatic participants after a 90-min walk in nature [5], we predicted a mean (SD) difference of 7 (4) ml (blood)/100g (tissue)/min in the ACC after HT compared with handiwork in participants with chronic LBP. With an α-risk of 5%, power of 90% and potential loss to follow-up of 20%, we sought to include 16 participants.

SAS 9.4 software was used for data analysis. Quantitative variables were described using the mean (SD) and [95%CI] or median (Q1, Q3). Qualitative variables were expressed as absolute and relative frequencies (n/N, %). The mean ACC perfusion after handiwork and HT and the mean change in ACC blood perfusion from baseline were compared between activities using a paired Student t test. We performed a sensitivity analysis using a mixed model with activity, period and sequence as fixed effects and subject as a random effect. The missing ACC perfusion value of one participant after the second activity was imputed by the median ACC perfusion after the first activity.

## Results

Sixteen participants were included between May 2021 and March 2022 (Fig 1): 14 (87.5%) were women, the mean (SD) age was 49.3 (9.5) years and baseline lumbar pain intensity on NRS was 45.1 (27.2)/100 points (Table 1).

The participants randomized to the HT-then-handiwork sequence had a longer current LBP episode duration, more previous LBP-related sick leaves, higher LBP intensity scores and lower mental health scores than the other group. The mean (SD) ACC perfusion was 49.6 (10.1) ml (blood)/100g (tissue)/min after the 2 handiwork sessions and 49.5 (9.9) after the 2 HT sessions (S6 Fig).

The mean rumination score for all participants was 33.3 (8.1)/60 after the 2 handiwork sessions and 34.0 (7.4)/60 after the 2 HT sessions. The mean catastrophizing score was 11.0 (4.0)/20 after the 2 handiwork sessions and 11.8 (4.1)/20 after the 2 HT sessions (Table 2). The mean ACC perfusion score after handiwork did not differ from that after HT (Table 2). The rumination and catastrophizing scores after handiwork and HT were also very close (Table 2).

The median (Q1, Q3) duration between the end of the activity and the MRI was 40.0 min (27.5, 65.0), range 15–90.

The mean ACC perfusion, rumination and catastrophizing scores decreased very slightly after both activities, with no between-activity difference in change (Table 3).

The complementary sensitivity analysis using a mixed model found no effect of sequence (p = 0.52), period (p = 0.45) or activity (p = 0.87) on ACC perfusion.

The mean LBP intensity decreased after handiwork (mean difference: 10.3 (24.5), [-3.3, 23.8]) and after HT, mean difference (9.6 (22.6), [-2.5, 21.6]) (Table 4).

Scores for all 6 items of the credibility/expectancy questionnaire were slightly higher for handiwork than for HT (Table 5).

Mean satisfaction level was 90 (18)% for handiwork and 89 (11)% for HT. Mean acceptability level was 92 (14)% for handiwork and 91 (12)% for HT.

No adverse event was reported.

## Discussion

In contrast with our hypothesis, ACC blood perfusion did not decrease after HT (S6 Fig) or handiwork.

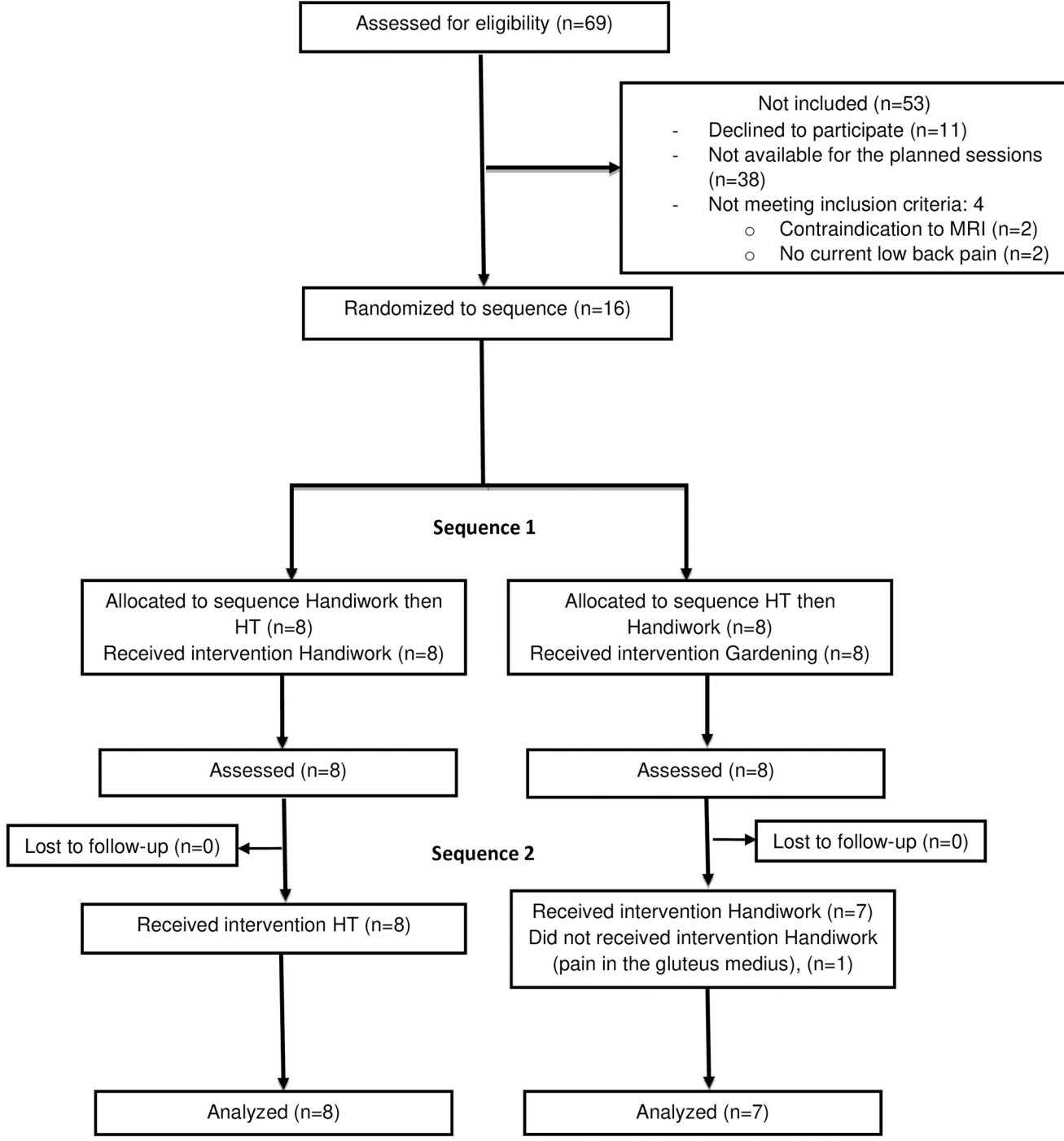

**Fig 1. Flow-chart.** HT: horticultural therapy.

The baseline mean ACC blood perfusion of the participants was close to the mean ACC blood perfusion measured with a PET scan in a study of asymptomatic participants [21], and the baseline rumination score was only slightly higher than that reported in young asymptomatic participants in the study by Bratman et al., (2015) [5]. Therefore, our participants may have been used to their long-lasting and stable symptoms, as shown by their moderate activity

**Table 1. Baseline characteristics of participants with chronic low back pain.**

| | Handiwork then Horticultural Therapy | Horticultural Therapy then Handiwork | All sequences and participants |
|---|---|---|---|
| | n = 8 | n = 8 | N = 16 |
| **Demographic characteristics** | | | |
| Age, mean (SD), years | 47.9 (10.0) | 50.8 (9.3) | 49.3 (9.5) |
| Female n (%) | 7 (87.5%) | 7 (87.5%) | 14 (87.5%) |
| Body mass index (kg.m$^{-2}$), mean (SD) | 22.6 (2.7) | 24.0 (2.8) | 23.3 (2.8) |
| Completed higher education, n (%) | 8 (100.0%) | 7 (87.5%) | 15 (93.8%) |
| **Clinical characteristics** | | | |
| Current LBP episode duration, mean (SD), months | 35.3 (9.5) | 67.0 (43.6) | 58.4 (39.6) |
| LBP intensity on NRS (range, 1–100), mean (SD) | 32.8 (24.0) | 57.5 (25.6) | 45.1 (27.2) |
| Radicular pain intensity on NRS (range, 1–100), mean (SD) | 12.5 (15.8) | 17.5 (26.6) | 15.0 (21.3) |
| Activity limitation: RMQD, score (0–24), mean (SD) | 7.9 (5.1) | 10.6 (2.2) | 9.3 (4.1) |
| Strategies for coping with pain: CSQ, catastrophizing sub-score (0–20) | 12.9 (3.6) | 13.6 (3.2) | 13.3 (3.3) |
| MOS SF-12 score, mean (SD) | | | |
| Physical component summary (range, 9.95–70.02) | 47.1 (10.2) | 42.8 (5.8) | 45.0 (8.3) |
| Mental component summary (range, 5.89–71.97) | 43.7 (16.2) | 38.0 (14.4) | 40.9 (15.1) |
| **Specific characteristics** | | | |
| ACC perfusion (ml (blood)/100g (tissue)/min)* mean (SD) | 48.2 (8.9) | 51.0 (14.9) | 49.5 (11.7) |
| Rumination score | 32.6 (6.7) | 35.9 (6.4) | 34.3 (6.5) |
| Gardening practice | | | |
| Never | 2 (25.0%) | 0 (0.0%) | 2 (12.5%) |
| Rarely (few times in the year) | 5 (62.5%) | 1 (12.5%) | 6 (37.5%) |
| Regularly (once or several times a month) | 1 (12.5%) | 6 (75.0%) | 7 (43.8%) |
| Often (once or several times a week) | 0 (0.0%) | 1 (12.5%) | 1 (6.3%) |
| **Main anatomical findings, n (%)** | | | |
| Degenerative disk disease | 4 (50.0%) | 6 (75.0%) | 10 (62.5%) |
| Active discopathy | 4 (50.0%) | 1 (12.5%) | 5 (31.3%) |
| Facet joint osteoarthritis | 2 (25.0%) | 6 (75.0%) | 8 (50.0%) |
| Spondylolisthesis | 0 (0.0%) | 1 (12.5%) | 1 (6.3%) |
| Trunk extensor weakness | 1 (12.5%) | 1 (12.5%) | 2 (12.5%) |
| Undertermined | 1 (12.5%) | 0 (0.0%) | 1 (6.3%) |

Abbreviations: LBP: low back pain; NRS, numerical rating scale, higher scores indicate higher pain intensity; RMQD: Roland-Morris Disability Questionnaire, higher scores indicate more limitation; CSQ: Coping Strategies Questionnaire (catastrophizing sub-score), higher scores indicate greater catastrophizing; MOS SF-12: Medical Outcomes Study Short Form 12, higher scores indicate better health; ACC: anterior cingulate cortex;

*n = 15/16.

The total number of pathologies exceeds the number of participants because one participant could have more than one etiology (anatomical finding) for LBP.

limitation scores. ACC activation has been associated with episodes of increasing pain in people with chronic LBP [22]. Another potential explanation for the similarity between the ACC blood perfusion and rumination scores found in our study of people with chronic LBP and the study of asymptomatic participants by Bratman et al. (2015) is that the asymptomatic participants may have had unassessed symptoms that could be associated with ACC activity and rumination, such as depression and anxiety [23–25].

The differences in the ACC blood flow scores between handiwork and HT and the change in ACC blood flow after both HT and handiwork was negligible; a change in cerebral blood

**Table 2. ACC perfusion after handiwork and horticultural therapy.**

| | Handiwork (n = 15) | Horticultural therapy (n = 15) | p-value | Mean differences [95%CI] |
|---|---|---|---|---|
| ACC perfusion (ml (blood)/100g (tissue)/min) | | | | |
| Mean (SD) | 49.7 (10.1) | 49.3 (9.9) | 0.80 | 0.4 |
| [95% CI] | [44.3, 55.10] | [44.0, 54.5] | | [-6.9, 7.7] |
| Rumination score | | | | |
| Mean (SD) | 33.3 (8.1) | 34.0 (7.4) | 0.99 | -7.0 |
| [95% CI] | [28.8, 37.7] | [30.0, 38.0] | | [-6.4, 5.0] |
| Catastrophizing score | | | | |
| Mean (SD) | 11.0 (4.0) | 11.8 (4.1) | 0.22 | -0.8 |
| [95% CI] | [8.8, 13.2] | [9.6, 14.0] | | [-3.8, 2.2] |

Rumination was assessed by the rumination sub-score of the Rumination Reflection Questionnaire, higher scores indicate a higher degree of rumination. Catastrophizing was assessed by the Coping Strategy Questionnaire sub-score, higher sub-scores indicate greater catastrophizing. n = 15/16, the missing ACC perfusion value of one participant after the second activity was imputed by the median ACC perfusion after the first activity.

flow (CBF) <5% may relate to measurement error [26]. The limited dose of "exposure" to nature, in terms of duration and immersion (green spaces in an urban setting) and/or the time between the intervention and the MRI may explain the lack of neurobiological effect. The physical activity involved by gardening may have counteracted the nature-related changes in ACC activity, in contrast with more passive nature exposures [5, 10]. The absence of an effect of sequence, period or activity on ACC perfusion may suggest stability in ACC blood flow, independent of challenging requirements [27].

Disparate results have been found between studies of nature-related changes in ACC activity. A study using task functional magnetic resonance imaging (fMRI) in 28 asymptomatic individuals showed greater activity in the ACC following exposure to nature scenic views compared to urban views [28]. Another study using resting-state fMRI to assess brain functional activity in 23 young asymptomatic individuals showed increased functional connectivity in several brain regions including the ACC, after participation in different horticultural tasks [29]. Therefore, the association between ACC activity and exposure to nature is not

**Table 3. Change in the primary and secondary outcomes from baseline.**

| | Handiwork (n = 15) | Horticultural therapy (n = 15) | p value | Mean differences [95%CI] |
|---|---|---|---|---|
| Change in ACC perfusion (ml (blood)/100g (tissue)/min) | | | | |
| Mean (SD) | -0.1 (10.7) | -0.1 (8.7) | 0.91 | 0.0 |
| [95% CI] | [-5.6, 5.8] | [-4.7, 4.6] | | [-6.9, 6.9] |
| Change in rumination score | | | | |
| Mean (SD) | -0.5 (4.4) | -0.3 (2.8) | 0.99 | -0.2 |
| [95% CI] | [-2.9, 2.0] | [-1.7, 1.2] | | [-2.8, 2.4] |
| Change in catastrophizing score | | | | |
| Mean (SD) | -2.0 (2.8) | -1.4 (2.3) | 0.22 | -0.6 |
| | IC95% [-3.5, -0.5] | IC95% [-2.7, -0.2] | | |
| [95% CI] | [-3.5, -0.5] | [-2.7, -0.2] | | [-2.4, 1.2] |

Rumination was assessed by the rumination sub-score of the Rumination Reflection Questionnaire, higher scores indicate higher degree of rumination; Catastrophizing was assessed by the Coping Strategy Questionnaire sub-score, higher sub-scores indicate greater catastrophizing. n = 15/16, the missing ACC perfusion value of one participant after the second activity was imputed by the median ACC perfusion after the first activity.

**Table 4. Change in the clinical characteristics of participants after handiwork and horticultural therapy.**

| | Baseline | Handiwork | Horticultural therapy |
|---|---|---|---|
| | (n = 16) | (n = 15) | (n = 16) |
| LBP intensity on NRS (1–100), mean (SD) | 45.1 (27.2) | 31.9 (21.6) | 35.6 (26.4) |
| Radicular pain intensity on NRS (1–100), mean (SD) | 15.0 (21.3) | 3.6 (6.5) | 11.4 (22.8) |
| Activity limitation: RMQD, score (0–24), mean (SD) | 9.3 (4.1) | 8.0 (4.1) | 8.3 (4.4) |
| MOS SF-12 score, mean (SD) | | | |
| Physical component summary (9.95–70.02) | 47.1 (10.2) | 45.6 (6.7) | 44.2 (8.1) |
| Mental component summary (5.89–71.97) | 43.7 (16.2) | 43.5 (9.6) | 43.9 (12.2) |
| HAD scale, anxiety score, (0–21), mean (SD) | 7.8 (3.1) | 7.4 (3.4) | 7.3 (3.3) |
| HAD scale, depression score (0–21), mean (SD) | 4.8 (3.5) | 4.7 (3.6) | 4.4 (3.8) |
| FABQ (work), (0–42), mean (SD) | 8.9 (9.4) | 7.0 (6.3) | 6.1 (8.5) |
| FABQ (physical activity) (0–24), mean (SD) | 6.6 (6.2) | 7.5 (6.0) | 6.4 (6.4) |

Abbreviations: LBP: low back pain; NRS, numerical rating scale, higher scores indicate greater pain; RMQD: Roland Morris Disability Questionnaire, higher scores indicate more limitation; CSQ: Coping Strategies Questionnaire (catastrophizing sub-score), higher scores indicate greater catastrophizing; MOS SF-12: Medical Outcomes Study Short Form 12, higher scores indicate better health; HAD: Hospital Anxiety and Depression scale, higher sub-scores indicate greater symptoms of anxiety/ depression; FABQ: Fear and Avoidance Beliefs Questionnaire, higher sub-scores indicate greater levels of fear and avoidance beliefs regarding work/physical activity.

unequivocal. Moreover, the affective motivational dimension of pain involves several other brain structures that we did not evaluate [30]; their activity might have been more modified than that of the ACC in our study.

The credibility/expectancy questionnaire scores might indicate that the participants thought and felt that HT was less effective than handiwork in improving their symptoms. Both handiwork and HT may have positive effects on LBP. The changes in other clinical scores after both activities were small.

Our study has several limitations. The clinical severity of chronic LBP differed between the activity sequence groups. The sample may have been too small to draw firm conclusions about the efficacy or absence of efficacy of HT on ACC activity. Some environmental variables that were not collected (weather conditions, stress, impact of the covid-19 pandemic etc.) could also have influenced ACC perfusion. Arterial spin labeling may not be sufficiently sensitive to detect changes in ACC perfusion.

**Table 5. Credibility/expectancy questionnaire (self-administered at the end of each activity).**

| | Handiwork | Horticultural therapy |
|---|---|---|
| | (n = 15) | (n = 16) |
| **Set 1** | | |
| Question 1 (0–9) At this point, how logical does the therapy offered to you seem? | 6.5 (1.6) | 6.1 (2.7) |
| Question 2 (0–9) At this point, how successfully do you think this therapy will be in reducing your symptoms? | 6.1 (2.0) | 5.4 (2.7) |
| Question 3 (0–9) How confident would you be in recommending this therapy to a friend who experiences similar problems? | 7.0 (2.7) | 6.8 (2.9) |
| Question 4 (%) By the end of the therapy, how much improvement in your symptoms do you think will occur? | 61.7 (14.6) | 58.1 (28.0) |
| **Set 2** | | |
| Question 1 (0–9) At this point, how much do you really feel that the therapy will help you to improve your symptoms? | 6.1 (1.8) | 4.9 (2.6) |
| Question 2 (%) By the end of the therapy, how much improvement in your symptoms do you really feel will occur? | 60.1 (14.3) | 52.6 (28.6) |

All results are expressed as mean (SD). Score: 0-9/0-100%; 0: no credibility or expectancy in the received therapy, 9/100%: maximal credibility or expectancy in the received therapy; Set 1 investigates of what the participant think about the treatment and the Set 2 what the participant feels about the treatment.

## Conclusion

Our results showed that ACC blood perfusion did not decrease after 2, 90-minute HT sessions in participants with chronic low back pain. Our study emphasizes the need for future studies to specify the exact effects of nature experiences, the optimal conditions of this experience for the most common diseases, and the most suitable outcomes to measure the impact of nature on different symptoms.

## Supporting information

**S1 File.**
(PDF)

**S1 Fig. Consolidated Standards of Reporting Trials (CONSORT) checklist.**
(PDF)

**S2 Fig. The Template for Intervention Description and Replication (TIDieR).**
(PDF)

**S3 Fig. Timeline sequence.**
(PDF)

**S4 Fig. Example of horticultural activity: 2 participants are pulling up wilted plants under the supervision of the occupational therapist (CB).**
(TIF)

**S5 Fig. Example of handiwork activity: The participant is sawing a wooden plank under the supervision of the occupational therapist (CB) (not seen on the picture).**
(TIF)

**S6 Fig. Average CBF maps in MNI152 space for the 16 participants before and after HT.** The pink outline corresponds to the anterior division of the cingulate gyrus of the Harvard-Oxford cortical structural atlases with a threshold at the probability of 50%. The quality of the image is linked to the low intrinsic resolution of the ASL.
(TIF)

## Acknowledgments

We thank Léa Jilet, Camille Ollivier and Sylvain Charron for the statistical analysis, and Nadjia Hamidi and Louisa Sidhoum from the Service de Rééducation et de Réadaptation de l'Appareil Locomoteur et des Pathologies du Rachis for their help in escorting participants from one hospital to another. We acknowledge Johanna Robertson, PT, PhD for professional copy editing.

## Author Contributions

**Conceptualization:** Alexandra Rören, Clement Debacker, François Rannou, Catherine Oppenheim, Christelle Nguyen.

**Formal analysis:** Alexandra Rören, Clement Debacker, Marc Saghiah, Hendy Abdoul.

**Funding acquisition:** Alexandra Rören.

**Investigation:** Clement Debacker, Marc Saghiah, Catherine Bedin, Christelle Nguyen.

**Methodology:** Alexandra Rören, Clement Debacker, Hendy Abdoul, Catherine Oppenheim, Christelle Nguyen.

**Resources:** Alexandra Rören, Clement Debacker, Marc Saghiah, Catherine Bedin, Anna Fayolle, Marie-Martine Lefèvre-Colau, François Rannou, Catherine Oppenheim, Christelle Nguyen.

**Software:** Clement Debacker, Marc Saghiah.

**Supervision:** Alexandra Rören, Christelle Nguyen.

**Writing – original draft:** Alexandra Rören, Clement Debacker, Christelle Nguyen.

**Writing – review & editing:** Alexandra Rören, Clement Debacker, Marc Saghiah, Catherine Bedin, Anna Fayolle, Hendy Abdoul, Marie-Martine Lefèvre-Colau, François Rannou, Catherine Oppenheim, Christelle Nguyen.

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
