## [Decision Letter · Decision Letter 0]

29 Aug 2024

PONE-D-24-15656Effects of horticultural therapy versus handiwork on anterior cingulate cortex activity in people with chronic low back pain: a randomized, controlled, cross-over, pilot studyPLOS ONE

Dear Dr. Roren,

Thank you for submitting your manuscript to PLOS ONE. After careful consideration, we feel that it has merit but does not fully meet PLOS ONE’s publication criteria as it currently stands. Therefore, we invite you to submit a revised version of the manuscript that addresses the points raised during the review process.

**Please note that we have only been able to secure a single reviewer to assess your manuscript. We are issuing a decision on your manuscript at this point to prevent further delays in the evaluation of your manuscript. Please be aware that the editor who handles your revised manuscript might find it necessary to invite additional reviewers to assess this work once the revised manuscript is submitted. However, we will aim to proceed on the basis of this single review if possible. **

We look forward to receiving your revised manuscript.

Kind regards,

Joanna Tindall, PhD

Staff Editor

PLOS ONE

**Journal Requirements:**

The study was funded by the Trophées patients AP-HP, the Fondation Roi Baudoin, (Fonds Luc), 2018 and the fonds MSERI hôpitaux AP-HP. Centre, Université Paris Cité 2019 and sponsored by the Département de la Recherche Clinique et du Développement de l’Assistance Publique-Hôpitaux de Paris. The funders had no role in the design and conduct of the study; collection, management, analysis, and interpretation of the data; preparation, review, or approval of the manuscript; and decision to submit the manuscript for publication.

We thank Léa Jilet, Camille Ollivier and Sylvain Charron for the statistical analysis, and Nadjia Hamidi and Louisa Sidhoum from the Service de Rééducation et de Réadaptation de l’Appareil Locomoteur et des Pathologies du Rachis for their help in escorting participants from one hospital to another. We acknowledge Johanna Robertson for professional copy editing.

AR

Fonds MSERI hôpitaux AP-HP. Centre, Université Paris Cité 2019.

AR

 Fondation Roi Baudoin, (Fonds Luc), 2018 Belgium

AR 

Trophées patients APHP

The funders had no role in the design and conduct of the study; collection, management, analysis, and interpretation of the data; preparation, review, or approval of the manuscript; and decision to submit the manuscript for publication.

Reviewers' comments:

Reviewer's Responses to Questions

**Comments to the Author**

1. Is the manuscript technically sound, and do the data support the conclusions?

Reviewer #1: Yes

2. Has the statistical analysis been performed appropriately and rigorously? 

Reviewer #1: No

3. Have the authors made all data underlying the findings in their manuscript fully available?

Reviewer #1: Yes

4. Is the manuscript presented in an intelligible fashion and written in standard English?

Reviewer #1: Yes

5. Review Comments to the Author

**Reviewer #1:** Recommedation

Major revision (although not difficult to do)

Although this pilot randomised study appears well designed it is perhaps too detailed for the patient numbers concerned (see Table 1). As the authors will know, patient numbers in pilot studies are likely to be insufficient to draw firm conclusions. The main focus is then to provide information for a possible future trial. In this respect, a key question is how large a difference between the two treatment arms is likely? Consequently, the information in Table 2 is important but lacks the magnitude of the differences between groups and the confidence interval on these differences. What is required in Table 2, for the ‘Change in catastrophizing score’ for example is, d = Han – Hor = −2.0 – (−1.4) = −0.6, 95% CI −2.4 to +1.2). [Authors please check the calculation]. From this the likely range of values one might plan for in a subsequent trial is indicated. The P-values given does not help with this.

I also wondered if a better analysis would compare post-intervention values of the Score (rather than the change). This might allow an easier interpretation of the results. The observed difference on this scale could then be checked for any influence of the corresponding baseline measures using regression methods.

I don’t think using a sensitivity analysis (lines 195-196) is useful here. I suggest the associated details are omitted.

6. PLOS authors have the option to publish the peer review history of their article (what does this mean?). If published, this will include your full peer review and any attached files.

Reviewer #1: No

---

## [Author Response · Author response to Decision Letter 0]

19 Oct 2024

Response to reviewer

We thank the Reviewer and the Editor for their appreciation of our manuscript. Below we respond to all the comments on a point-to-point basis.

This pilot randomised study appears well designed it is perhaps too detailed for the patient numbers concerned (see Table 1). As the authors will know, patient numbers in pilot studies are likely to be insufficient to draw firm conclusions.

In accordance with this remark, we have now simplified Table 1 and retained only essential information.

We agree that our sample size may be not sufficient to draw firm conclusion. This point was already outlined in the limitations paragraph, it is now better specified

“The sample may have been too small to draw firm conclusions about the efficacy or absence of efficacy of HT on ACC activity”. L. 275-6

In this respect, a key question is how large a difference between the two treatment arms is likely? Consequently, the information in Table 2 is important but lacks the magnitude of the differences between groups and the confidence interval on these differences. What is required in Table 2, for the ‘Change in catastrophizing score’ for example is, d = Han – Hor = −2.0 – (−1.4) = −0.6, 95% CI −2.4 to +1.2). [Authors please check the calculation]. 

Thank you for this important point. We have now added the magnitude of the differences between groups by estimating the differences in mean and their confidence intervals (See new Tables 2 and 3). 

I also wondered if a better analysis would compare post-intervention values of the Score (rather than the change). This might allow an easier interpretation of the results. The observed difference on this scale could then be checked for any influence of the corresponding baseline measures using regression methods.

Thank you for this pertinent suggestion. We have added a table (Table 2) reporting the comparison of the ACC, rumination and catastrophizing values after horticultural therapy and after handiwork. These results show no differences in ACC perfusion between HT and handiwork and slight differences from baseline as explained in the following sentences

“The mean ACC perfusion score after handiwork did not differ from that after HT (Table 2). The rumination and catastrophizing scores after handiwork and HT were also very close (Table 2).” L. 209-11.

We did not run any models, because we did not find any reason for isolating demographic or clinical factor(s) that would explain the results.

I don’t think using a sensitivity analysis (lines 195-196) is useful here. I suggest the associated details are omitted.

We understand your point of view, nevertheless, we preferred to keep the sensitivity analysis because it was planned in order to explore the potential confounding factors. The results of the sensitivity analysis showed that there was no association between sequence, period or activity and ACC perfusion. We added a sentence in the discussion to illustrate the information brought by the sensitivity analysis.

“The absence of effect of sequence, period or activity on ACC perfusion may suggest stability in ACC blood flow, independent of active requirements [27]”. L. 256-8

---

## [Decision Letter · Decision Letter 1]

4 Nov 2024

Effects of horticultural therapy versus handiwork on anterior cingulate cortex activity in people with chronic low back pain: a randomized, controlled, cross-over, pilot study

PONE-D-24-15656R1

Dear Dr. Roren,

We’re pleased to inform you that your manuscript has been judged scientifically suitable for publication and will be formally accepted for publication once it meets all outstanding technical requirements.

Kind regards,

Stephen D. Ginsberg, Ph.D.

Section Editor

PLOS ONE

Reviewers' comments:

Reviewer's Responses to Questions

**Comments to the Author**

1. If the authors have adequately addressed your comments raised in a previous round of review and you feel that this manuscript is now acceptable for publication, you may indicate that here to bypass the “Comments to the Author” section, enter your conflict of interest statement in the “Confidential to Editor” section, and submit your "Accept" recommendation.

Reviewer #1: All comments have been addressed

2. Is the manuscript technically sound, and do the data support the conclusions?

Reviewer #1: (No Response)

3. Has the statistical analysis been performed appropriately and rigorously? 

Reviewer #1: (No Response)

4. Have the authors made all data underlying the findings in their manuscript fully available?

Reviewer #1: (No Response)

5. Is the manuscript presented in an intelligible fashion and written in standard English?

Reviewer #1: (No Response)

6. Review Comments to the Author

Reviewer #1: The authors have made changes to the paper following suggestions made in my earlier review I have no further comments.

7. PLOS authors have the option to publish the peer review history of their article (what does this mean?). If published, this will include your full peer review and any attached files.

Reviewer #1: No

---

## [Editor Report · Acceptance letter]

28 Nov 2024

PONE-D-24-15656R1 

PLOS ONE

Dear Dr. Rören, 

I'm pleased to inform you that your manuscript has been deemed suitable for publication in PLOS ONE. Congratulations! Your manuscript is now being handed over to our production team.

Kind regards, 

on behalf of

Dr. Stephen D. Ginsberg 

Section Editor

PLOS ONE